# Excited-State Dynamics of All the Mono-*cis* and the Major Di-*cis* Isomers of β-Apo-8′-Carotenal as Revealed by Femtosecond Time-Resolved Transient Absorption Spectroscopy

**DOI:** 10.3390/molecules28114424

**Published:** 2023-05-29

**Authors:** Kota Horiuchi, Chiasa Uragami, Ruohan Tao, Daisuke Kosumi, Richard J. Cogdell, Hideki Hashimoto

**Affiliations:** 1Department of Applied Chemistry for Environment, Graduate School of Science and Technology, Kwansei Gakuin University, 1 Gakuen-Uegahara, Sanda 669-1330, Japan; 2Institute of Industrial Nanomaterials, Kumamoto University, 2-39-1 Kurokami, Kumamoto 860-8555, Japan; 3School of Molecular Biosciences, College of Medical, Veterinary and Life Sciences, University of Glasgow, Glasgow G12 8QQ, UK

**Keywords:** carotenoid, *cis* isomer, ICT excited state, energy-gap law, ultrafast spectroscopy, photoisomerization

## Abstract

*Cis* isomers of carotenoids play important roles in light harvesting and photoprotection in photosynthetic bacteria, such as the reaction center in purple bacteria and the photosynthetic apparatus in cyanobacteria. Carotenoids containing carbonyl groups are involved in efficient energy transfer to chlorophyll in light-harvesting complexes, and their intramolecular charge–transfer (ICT) excited states are known to be important for this process. Previous studies, using ultrafast laser spectroscopy, have focused on the central-*cis* isomer of carbonyl-containing carotenoids, revealing that the ICT excited state is stabilized in polar environments. However, the relationship between the *cis* isomer structure and the ICT excited state has remained unresolved. In this study, we performed steady-state absorption and femtosecond time-resolved absorption spectroscopy on nine geometric isomers (7-*cis*, 9-*cis*, 13-*cis*, 15-*cis*, 13′-*cis*, 9,13′-*cis*, 9,13-*cis*, 13,13′-*cis*, and all-*trans*) of β-apo-8′-carotenal, whose structures are well-defined, and discovered correlations between the decay rate constant of the S_1_ excited state and the S_0_−S_1_ energy gap, as well as between the position of the *cis*-bend and the degree of stabilization of the ICT excited state. Our results demonstrate that the ICT excited state is stabilized in polar environments in *cis* isomers of carbonyl-containing carotenoids and suggest that the position of the *cis*-bend plays an important role in the stabilization of the excited state.

## 1. Introduction

Photosynthesis relies on the light-harvesting capabilities of chlorophylls, which mainly absorb light energy in the ultraviolet (Soret absorption band) and far-red/near-infrared (Q_y_ absorption band) regions of the solar spectrum. However, chlorophylls are unable to efficiently absorb blue to green light [1]. To bridge this spectral gap, carotenoids are utilized to enhance photosynthetic light-harvesting [2]. Carotenoids can capture blue to green light and transfer that absorbed energy to chlorophylls via singlet-singlet energy transfer. The efficiency of this process depends on the type of carotenoid present in the pigment-protein complexes of photosynthetic organisms [3]. Carbonyl-containing carotenoids are particularly effective at energy transfer, due to the presence of carbonyl groups that are in conjugation with their polyene backbone [3]. However, the molecular mechanisms behind their high efficiency in this energy transfer process are still debated [4,5,6,7,8,9,10,11]. A detailed understanding of the excited-state dynamics of carbonyl-containing carotenoids will be important for developing highly efficient, energy transfer systems for artificial photosynthesis [12,13].

The polyene conjugated system of linear carotenoid molecules is often approximated as belonging to the C_2h_ point group symmetry [2,14,15]. However, the singlet excited-states of carotenoid show an unusual pattern of the state ordering due to the strong electron–electron interaction of the π-electrons in the conjugation system [2]. When light is absorbed by a carotenoid, the transition from the ground singlet (S_0_, 1^1^A_g_^−^) state to the second excited singlet (S_2_, 1^1^B_u_^+^) state is allowed in a one-photon optical transition. On the other hand, the transition from the S_0_ ground state to the first excited singlet (S_1_, 2^1^A_g_^−^) state is one-photon forbidden. Moreover, ultrafast time-resolved laser spectroscopy has confirmed the presence of additional excited states designated as intramolecular charge-transfer (ICT) and S* excited states [2,3]. These additional excited states play important roles, respectively, in the light-harvesting and photo-protection, creating a more detailed insight into the generation mechanisms of these eccentric excited states, which are essential to understanding the excited-state dynamics of carotenoids. Especially, the ICT excited state of the carbonyl-containing carotenoid is important because this excited state is stabilized in the polar environment and facilitates the highly efficient singlet–singlet excitation energy transfer from carotenoid to chlorophyll in the light-harvesting systems of diatoms and marine algae [3,16]. Although there are multiple explanations for the generation mechanisms of the ICT excited state [4,5,6,7,8,9,10,11], the ICT state is often attributed to a mixed S_1_/ICT state due to its strong coupling with the S_1_ state. Previous studies have investigated ICT excited states from the viewpoint of their dependencies on the solvent [9,17,18,19,20], conjugation length [7,8,18,21], excitation wavelength [22] and substituent groups [6,7]. In addition, the ultrafast relaxation process after photoexcitation of the geometric isomers of carbonyl-containing carotenoids has also been investigated. However, the number of reports is restricted and reliable data on the integrity of the isomers as well as the isomerization following photoexcitation are sparse [14,23]. Nonetheless, the isomers of carotenoids hold significant importance in the photosynthetic reaction center (RC), and thus far, the subsequent facts have been definitively established. Carotenoids bound to the RC of purple photosynthetic bacteria adopt a 15-*cis* configuration of the carotenoid [24,25]. In the cyanobacterial photosystem I (PSI), there are 22 β-carotenes, of which, 17 are all-*trans*, 2 are 9-*cis*, and each one comprises 9,9′-*cis*, 9,13′-*cis*, 13-*cis*, and they act as both light-harvesters and photoprotectors [26]. Thus, the *cis* isomers of carotenoids are certainly utilized in the primary process of photosynthesis, but the fundamental question of why the *cis* isomers, rather than the all-*trans* isomers, are selected, still needs addressing.

Another example of an unexplained excited state is the S* state. The shoulder of the transient-absorption band appearing on the high-energy side of the S_1_ → S_n_ transition of carotenoids at room temperature has been assigned to the S* state. The S* state generated in carotenoids bound to antenna pigment-protein complexes has been postulated to be the precursor of the triplet excited states of the carotenoids [2,27,28]. On the other hand, the S* state generated in carotenoids in organic solvents was claimed, on the basis of pump-deplete-probe spectroscopy, due to the vibrationally hot ground state that is formed by both the S_1_ → S_0_ internal conversion, and the impulsive Raman scattering promoted by the pump-pulse irradiation [27]. What is clear is that the origin of the S* state is still under debate, and the exact generation process remains to be clarified.

The excited-state dynamics of the *cis* isomers of peridinin and fucoxanthin, both of which are well-known carbonyl-containing carotenoids, have been investigated [14,23]. In these studies, it was reported that the *cis* isomers in polar solvents exhibited transient absorption bands characteristic of the ICT excited state. The shortening of the S_1_/ICT lifetimes was also reported, compared to the S_1_ lifetimes in nonpolar solvents. In addition, the *cis* isomer had a shorter S_1_ or S_1_/ICT lifetime than that of the all-*trans* isomer. It should be noted that the shortening of the S_1_ lifetimes of the *cis* isomers has also been observed in carotenoids without carbonyl groups [29,30]. Previous studies on *cis* isomers of carotenoids have primarily focused on central-*cis* isomers, which exhibit a *cis*-bend at the central portion of the polyene chain. However, there has been insufficient research on terminal-*cis* isomers and di-*cis* isomers, and the integrity of the *cis*-configuration has often not been well established. In order to bridge these knowledge gaps, our research focuses on exploring the isomers of the carbonyl-containing carotenoid found in β-apo-8′-carotenal. Hashimoto et al. [31] have already isolated and identified all the mono-*cis* and major di-*cis* isomers of this carotenoid, and their chemical structures can be seen in Figure 1. Using femtosecond time-resolved absorption spectroscopy, we have studied the excited-state dynamics of these isomers in both non-polar and polar solvents. We hypothesize that the asymmetry of the molecule and intentional breaking of the C_2h_ symmetry, with the introduction of the *cis*-bend, may impact the electronic structure of the linear polyene.

## 2. Results and Discussion

### 2.1. HPLC Profiles and Steady-State Absorption Spectra of Isomeric β-Apo-8′-Carotenal

Figure 2 shows the HPLC profiles of the isomers of β-apo-8′-carotenal. The assignment of each peak was performed by reference to our previous report [31], and the result is summarized in Table 1. The 0-2, 0-1, 0-0 absorption wavelengths of the main S_0_ → S_2_ absorption band, as well as the 0-1, 0-0 absorption wavelengths of the *cis* peak (S_0_ → S_3_ absorption; see below) of each isomer, which were determined in *n*-hexane solution at room temperature, are also shown in Table 1. The peak wavelengths of these sub-bands (vibrational structure) were determined by calculating the second-derivative waveforms of the absorption spectra of the isomers (see Appendix A). The 0-1 and 0-0 absorption wavelengths of the S_0_ → S_2_ transition of all the *cis* isomers show a blue shift compared to that of the all-*trans* isomer. The extent of the shift is smaller than 8 nm for the mono-*cis* isomers, while that of the di-*cis* isomers are larger than 9 nm. In contrast, the 0-0 absorption wavelengths of the *cis*-peaks are similar among the isomers and appear around 331 nm, although the 9-*cis* and 13,13′-*cis* isomers show slightly larger deviation from this mean value. These trends are similar to that of isomeric β-carotene, which were reported long ago [32]. The structural difference between β-apo-8′-carotenal and β-carotene is the presence or absence of the carbonyl group or β-ionone ring at one terminal end of the polyene chain. However, all the results shown above suggest that the shift of the main S_0_ → S_2_ absorption bands of the *cis* isomers of β-apo-8′-carotenal, with respect to the all-*trans* molecule in non-polar *n*-hexane solution, obey the rules that have been found previously for β-carotene.

Cerón-Carrasco et al. theoretically investigated the excited-states’ properties of the all-*trans*, all the mono-*cis*, and six di-*cis* isomers of β-carotene, using the DFT/MRCI method with the B3LYP/6-31G(d) basis set [33]. They claimed that the DFT/MRCI method is a strong choice to predict the absorption spectra of isomeric β-carotene, based on a comparison of the results with the TD-DFT(B3LYP/6-31G(d)) and TD-DFT/TDA (B3LYP/6-31G(d)) methods. It was found that there was a linear relationship between the C6-C6′ distance of the geometrically optimized isomers of β-carotene, and the theoretically predicted oscillator strengths of the S_0_ → S_2_ transition or *cis*-band (S_0_ → S_3_) transition. This finding demonstrates that the intensity ratio of the *cis* absorption band versus the main absorption band can be explained for by taking into account the two-dimensional molecular framework of isomeric β-carotene. Namely, the length of the long molecular axis (C6-C6′) can be a strong measure to use as a comparison with the intensity ratio of the *cis* band/main band absorptions among the isomers, since it is expected that the transition dipole moment of the main S_0_ → S_2_ absorption is proportional to the C6-C6′ length, while that of the *cis*-band is expected to be inversely proportional to the C6-C6′ length. This implies that the *cis* isomers exhibiting a longer C6-C6′ length should, correspondingly, have a shorter length along the shorter molecular axis, which is directly proportional to the transition dipole moment for *cis* absorption (and vice versa). We have applied a similar analysis to our set of isomers of β-apo-8′-carotenal, and the results are shown in Figure 3. To this end, the geometrically optimized structures of the isomers of β-apo-8′-carotenal in vacuo were determined using DFT calculations with the B3LYP/6-31G(d) basis set. Since β-apo-8′-carotenal lack a β-ionone ring at one terminal end, in comparison to β-carotene, we have adopted the C6-C8′ distance to represent the length of the long-molecular axis instead of the C6-C6′ distance, as has been adopted for β-carotene. As illustrated in Figure 3, both the isomers of β-carotene and β-apo-8′-carotenal show a strong linear relationship between the length of long-molecular axis (C6-C6′ distance for β-carotene and C6-C8′ distance for β-apo-8′-carotenal) and the ratio of theoretically or experimentally determined intensity of the *cis*-band transition divided by that of the S_0_ → S_2_ transition. This finding once again supports the idea that the absorption behavior of the isomers of β-apo-8′-carotenal in *n*-hexane obey the same rule that was found for the isomers of β-carotene. Namely, the present finding suggests that the S_0_ → S_2_ absorption behavior of the *cis* isomers of β-apo-8′-carotenal, whose chemical structures are asymmetric, can be well accounted for by referring to the results of the isomers of the symmetric carotenoid, β-carotene.

The observed similar trend between β-Apo-8′-carotenal and β-carotene molecules, despite the change in distance between carbons from 6-6′ to 6-8′, raises an interesting point. While these two substances have distinct terminal structures, it is intriguing that the same analysis method can be applied to both. One possible explanation for this phenomenon could be the presence of common structural features or functional groups that are responsible for the observed trend. Despite the differing terminal structures at one end, there may be underlying similarities in molecular properties or interactions that affect the analyzed behavior. These similarities could be manifested in the molecules’ responses to the employed analysis method. It is important to consider the specific analysis method used in the study, as it may focus on certain aspects of behavior or properties that are not solely determined by the terminal structures. For example, the analysis method might primarily assess electronic transitions, conjugation lengths, or molecular conformations, which could be influenced by factors beyond the terminal structures. Further investigation into the specific analysis method and a comparison of the molecular properties of β-Apo-8′-carotenal and β-carotene would provide valuable insights into how these distinct substances exhibit similar trends. By identifying the common factors contributing to the observed trend, researchers can obtain a better understanding of the underlying mechanisms, and potentially extend the application of the analysis method to other related molecules or systems.

We further investigated the solvent dependence of the absorption spectra of the isomers of β-apo-8′-carotenal. As an example, Figure 4 shows the absorption spectra of the all-*trans*, 15-*cis* and 9,13′-*cis* isomers of β-apo-8′-carotenal in *n*-hexane (non-polar solvent), acetone (polar aprotic solvent), and methanol (polar protic solvent) at room temperature (see Appendix A for the absorption spectra of all the isomers). In all the cases of the isomers, the main S_0_ → S_2_ absorption bands become asymmetrically broad in polar solvents and the vibrational structures become blurred. The extent of the broadening is stronger in methanol than in acetone, which may reflect the possibility of the hydrogen bonding of methanol to the C=O group of β-apo-8′-carotenal, in addition to the higher polarity of methanol than acetone. The *cis*-bands of the isomers show a red-shift in polar solvents, which is larger in methanol than in acetone. However, the linear relationship between the intensity ratio of *cis*-band/S_0_ → S_2_ transitions, which is similar to that in *n*-hexane, can also be seen in the cases of acetone and methanol solutions (see Appendix A). This suggests that the C6-C8′ distance can also be a strong measure to use in comparison with the intensity ratio of the *cis*-band/S_0_ → S_2_ transitions, even in the case of the isomers of β-apo-8′-carotenal in polar solvents.

### 2.2. Femtosecond Time-Resolved Absorption Spectroscopy

Figure 5 shows experimentally observed, femtosecond time-resolved absorption spectra of the all-*trans* and 15-*cis* isomers of β-apo-8′-carotenal recorded at selective delay times after excitation in *n*-hexane, acetone, and methanol (The experimental data of the other isomers are shown in Appendix A). In *n*-hexane solutions, both isomers show infra-red transient absorption bands that are ascribable to the S_2_ → S_m_ transitions that appear immediately after excitation, and then they transform into the transient absorption that is ascribable to the S_1_ → S_n_ transitions in the visible spectral region (here, S_m_ and S_n_ stand for the higher lying singlet excited states). The 15-*cis* isomer shows a broader feature for the S_1_ → S_n_ absorption, which is also seen with the other mono-*cis* and di-*cis* isomers (see Appendix A). In polar acetone solutions, the visible transient absorptions show an additional feature in the 600–700 nm wavelength region, which is ascribable to the production of the ICT excited state that is coupled to the S_1_ state. The ICT transient absorption band appears more strongly in polar protic methanol, which shows a strong agreement with the previous studies for the all-*trans* isomer [9,19]. It should be noted that the similarity between the components in the 600–700 nm region in non-polar (*n*-hexane) and polar (acetone) solvents can be observed for the 15-*cis* isomer, as shown in the top and middle panels of Figure 5b. This pattern was also reproduced by the global analysis, as indicated in Figure 6b. However, the spectral feature of the ICT transient absorption band is greatly enhanced in polar protic methanol, even in the case of the 15-*cis* isomer (see the bottom panel of Figure 5b).

To obtain a much deeper insight into the transient absorption feature, we have applied global analysis to the entire observed datasets of the time-resolved absorption spectra of the isomers β-apo-8′-carotenal. The results of the analysis for the all-*trans* and 15-*cis* isomers are shown in Figure 6 (see Appendix A for the results of the other isomers). All the time-resolved absorption datasets were well fitted by global analysis, using a three-component sequential model. The upper panel in Figure 6a shows the EADS (evolution associated difference absorption spectra) of the transient species produced by the excitation of the all-*trans* isomer in *n*-hexane. The first 90 ± 10 fs component can be assigned to the S_2_ state due to the bleaching and/or stimulated emission in the spectral region of the ground-state absorption (450–600 nm), and the subsequent transient absorption that is ascribable to the S_2_ → S_m_ transition in the 800–1000 nm spectral region. The second 540 ± 10 fs component can be assigned to the vibrationally hot S_1_ state, and the third 23.81 ± 0.02 ps component can be assigned to the relaxed S_1_ state. The third EADS shows the sharp transient absorption that is characteristic of the S_1_ → S_n_ transition. These results show a strong agreement with a previous report [9]. Similar results are observed for the 15-*cis* isomer, as illustrated in the upper panel in Figure 6b, although the first 110 ± 10 fs and second 800 ± 10 fs EADS show longer lifetimes compared to those of the all-*trans* isomer, and the third 18.70 ± 0.01 ps EADS has a shorter lifetime and a broader spectral feature than those of the all-*trans* isomer.

The middle panel in Figure 6a shows the EADS of the transient species produced by the excitation of the all-*trans* isomer in acetone. The time-resolved absorption dataset of the all-*trans* isomer in acetone can be well accounted for with three components: the first 120 ± 10 fs, the second 610 ± 10 fs, and the third 15.68 ± 0.01 ps components. In contrast to the results, in *n*-hexane solution, the sharp peak that is ascribable to the S_1_ → S_n_ transition around 550 nm becomes broader and enhancement of the broad absorption in the 600–700 nm spectral region is seen. This broad absorption feature in the 600–700 nm region is characteristic of the ICT state. Therefore, we tentatively assign the transient absorption band that gives rise to the third EADS (15.68 ± 0.01 ps component) to the S_1_/ICT states, since the attribution of specific features to S_1_ or ICT-photoinduced absorption is still under debate [2,3]. It is noted here that the first and the second EADS show the longer lifetimes than those in *n*-hexane, and the third EADS shows a shorter lifetime than that in *n*-hexane. Similar results are observed for the 15-*cis* isomer as illustrated in the middle panel in Figure 6b, although once again, the first 130 ± 10 fs and second 1.25 ± 0.01 ps EADS show longer lifetimes compared to those of the all-*trans* isomer in acetone, and the third 12.87 ± 0.01 ps EADS show a shorter lifetime than that of the all-*trans* isomer in acetone.

The bottom panel in Figure 6a shows the EADS of the transient species produced by the excitation of the all-*trans* isomer in methanol. The time-resolved absorption dataset of the all-*trans* isomer in methanol can be well accounted for with three components: the first 180 ± 10 fs, the second 1.89 ± 0.01 ps, and the third 6.10 ± 0.01 ps. The first and the second EADS show longer lifetimes than that in acetone and the third EADS shows a shorter lifetime than that in acetone. The ICT transient absorption band that appears in the 600–700 nm region is more pronounced in methanol than in acetone (see the third EADS in the bottom panel of Figure 6a). These results are in strong agreement with the previous report [9]. Similar results are observed for the 15-*cis* isomer in methanol, as illustrated in the bottom panel in Figure 6b, although once again the first 140 ± 10 fs and second 1.54 ± 0.01 ps EADS show longer lifetimes compared to those of the all-*trans* isomer in methanol, and the third 7.00 ± 0.03 ps EADS show a shorter lifetime than that of the all-*trans* isomer in methanol.

In order to obtain a clearer understanding of the underlying optical processes, Figure 7 represents the potential energy surfaces summarizing the excited-state absorption, internal conversion, and vibrational relaxation after the S_0_ → S_2_ excitation of β-apo-8′-carotenal. The red arrows indicate excited-state absorptions: S_1_ → S_n_, S_2_ → S_m_, S_1_ → S_3_, and S_1_/ICT → S_n_. The black, wavy lines represent internal conversion and vibrational relaxation in a non-polar solvent. In a polar-solvent environment, the potential energy surface of the S_1_/ICT state, shown in blue, is stabilized. Therefore, the vibrational relaxation and internal conversion in a polar solvent follow the path indicated by the blue, wavy lines. The potential energy surfaces of the S_n_ and S_m_ states are represented by thick, gray lines due to the lack of knowledge.

As illustrated in Appendix A, the spectral features of the EADS of all the other isomers show similar trends, as seen in the case of the 15-*cis* isomer except for the 7-*cis* isomer. Table 2 summarizes the wavelengths of the absorption maxima of the S_1_ → S_n_ and S_1_/ICT → S_n_ transitions of all the isomers. It was found that the maxima of the S_1_ → S_n_ transitions of the *cis* isomers, except for the 7-*cis* isomer, are red-shifted compared to the all-*trans* isomer (see Table 2). The extent of the red-shift of the di-*cis* isomers was somewhat larger than those of the mono-*cis* isomers. It is interesting to note that only in the case of the 7-*cis* isomer was a blue-shift of the transient absorption seen. Additionally, the S_1_ → S_n_ transient absorption of the 7-*cis* isomer in *n*-hexane is very different from those of the other isomers. This interesting feature is further discussed in detail below.

Figure 8a shows the EADS of the S_1_ species in *n*-hexane of the all-*trans* and all the mono-*cis* isomers of β-apo-8′-carotenal in the 460–600 nm spectral region. The S_1_ species of the 7-*cis* isomer in *n*-hexane shows a maximum at 520 nm, which is largely blue-shifted (−27 nm) from that of the all-*trans* isomer (see also Table 2). The EADS of the 7-*cis* isomer additionally shows a shoulder at 542 nm that lies on the longer wavelength-side from the maximum, while the other isomers show this shoulder on the shorter wavelength-side of the maximum. Additionally, the 542 nm shoulder of the 7-*cis* isomer is reminiscent of the main absorption band peaking around 560 nm of the other isomers. Indeed, the peak position of this shoulder of the isomers, except 7-*cis,* lies around 520 nm, which is in strong agreement with the main band of the 7-*cis* isomer peaking at 520 nm. The intensities of the 520 nm bands of the *cis* isomers are in the order: 7-*cis* >> 9-*cis* > 13-*cis* > 15-*cis* > 13′-*cis*. This order shows an accidental correspondence with the distance between the *cis*-bend and carbonyl oxygen. The 7-*cis* isomer has the longest distance, while the 13′-*cis* isomer has the shortest distance. As illustrated in Figure 8b, it is interesting to note that the intensity ratios of the 520 nm absorption, to around the 560 nm absorption in the EADS of the isomers shown in Figure 8a, show a very good linear relationship with the number of C=C bonds (*n_cis_*) from the C=O group to the *cis*-bend in the *cis* isomers, except for the 7-*cis* isomer. The large deviation from this trend is found only for the 7-*cis* isomer. This large deviation might be caused by the non-planar structure of the 7-cis isomer, due to steric repulsion between the 5-methyl group in the β-ionone ring and the 9-methyl group attached to the polyene backbone. This interesting spectral feature is somewhat maintained in the presence of acetone but is not seen in the presence of methanol. It should be noted here that the 520 nm absorption band is not due to the generation of the S* state [2,3]. This is because intensities of both the 520 nm and around 560 nm absorption bands show exactly the same temporal behavior. In our study, the lower energy edge of the S_0_ → S_2_ absorption was excited to produce the excited-state species. In contrast, it is known that excitation of the blue-side of the S_0_ → S_2_ absorption is required to generate the S* state [2,3].

### 2.3. The Relationship between the Lifetimes and the Energies of the S_1_ States of the Isomers in n-Hexane

It is known that the lifetimes of the S_1_ excited-state of all-*trans* carotenoids can be predicted theoretically using the energy-gap law for radiationless transitions as described by Engelman and Jortner [34,35] as shown by Equations (1) and (2) below.
(1)$$k=\frac{C^{2}{\left(2\pi \right)}^{\frac{1}{2}}}{\hbar {\left(\Delta E\hbar \omega_{M}\right)}^{\frac{1}{2}}}\exp\left(-\gamma \frac{\Delta E}{\hbar \omega_{M}}\right),$$
(2)$$\gamma =\ln\left(\frac{2\Delta E}{d\Delta_{M}^{2}\hbar \omega_{M}}\right)-1,$$

Here, k is the rate of the radiationless transition, ΔE is the energy gap between the S_1_ and S_0_ states of the all-*trans* carotenoids, C is the vibronic coupling matrix element, d is the number of degenerate or nearly degenerate modes of the frequency $\omega_{M}$, ${∆}_{M}$ is the reduced displacement of the accepting mode, and $\hbar \omega_{M}$ is the energy of the accepting vibrational modes.

Chynwat and Frank were successful to explain the relationship between the lifetimes and the energies of the S_1_ states of a series of all-*trans* carotenoids that had been determined experimentally by applying the energy-gap law [34]. On the other hand, Niedzwiedzki et al. claimed that some *cis* isomers do not obey the energy-gap law that has been established for the all-*trans* isomer [29]. This is because the observed lifetimes of the S_1_ species produced from the *cis* isomers were short compared to those predicted from the energy gap between the S_1_ and S_0_ states. The reason for this discrepancy was explained as being due to the electronic coupling terms (C in Equation (1)) that are significantly higher for the *cis* isomer, based on the quantum chemical calculations, and when combined with the Franck–Condon factors, the predicted internal conversion rates become roughly double of the all-*trans* species. To address this interesting issue once again, we have applied a similar analysis to the present complete set of the *cis*-*trans* isomers of β-apo-8′-carotenal in *n*-hexane.

In order to apply the energy-gap law to the present dataset, the S_1_ state energies (ΔE) of the *cis* isomers of β-apo-8′-carotenal in *n*-hexane must be determined. To do this, we have estimated the S_1_ state energies of the *cis* isomers by referring to the energies of the *cis*-peak in the steady-state absorption spectra and to the small absorption peaks that are seen in the 650–700 nm spectral region of the EADS of the S_1_ species of the *cis* isomers in *n*-hexane (see Appendix A). Since the *cis*-peak in the steady-state absorption is attributed to the S_0_ → S_3_ (1^1^A_g_^+^) transition and the small absorption peak that appears in the longer wavelength side of the main absorption band in the EADS of the S_1_ species is attributed to the S_1_ → S_3_ transition of the *cis* isomer [29,30], the S_1_ state energies can be calculated by subtracting the energies of the S_1_ → S_3_ transitions from those of the S_0_ → S_3_ transitions of the *cis* isomer in *n*-hexane. For the S_1_ energy of the all-*trans* isomer, the reported value (15,200 cm^−1^) was adopted [21].

Figure 9 shows the relationship between the relaxation rates and the S_0_–S_1_ energy gaps (ΔE) of all the isomers of β-apo-8′-carotenal that were determined in this study, and that of the reported all-*trans* carotenoids [36]. The data point for the all-*trans* isomer of β-apo-8′-carotenal (see solid purple circle in Figure 9) is on the line derived from the energy-gap law for the series of all-*trans* carotenoids. The data points of the other isomers of β-apo-8′-carotenal approximately follow the same line derived from the energy-gap law for the all-*trans* carotenoids, although some deviations are seen, as shown in the inset of Figure 9, which shows a strong agreement with the previous study [29]. It is interesting to note, however, that the data points of the *cis* isomers of β-apo-8′-carotenal can be classified into three groups: Group I, the *cis* isomers that contain 13- or 15-*cis* bend (solid red circles in Figure 9); Group II, the *cis* isomers that contain 13′- or 9-*cis* bend (solid dark-blue circles in Figure 9); and Group III, 7-*cis* isomer (solid green circle in Figure 9). By referring to the interpretation in the previous study [29], the *cis* isomers belonging to Group I show a shorter S_1_ lifetime than that of the all-*trans* isomer, which might be due to the fact that the electronic coupling terms (C in Equation (1)) are higher for these *cis* isomers. On the other hand, the *cis* isomers belonging to Group II show the similar S_1_ lifetimes with that of the all-*trans* isomer. Only in the case of the 7-*cis* isomer (Group III), a longer S_1_ lifetime can be seen than that of the all-*trans* isomer, suggesting that the electronic coupling term (C in Equation (1)) is lower for the case of 7-*cis* isomer.

In order to discuss this interesting issue more precisely, we can define a new parameter, *k_cis_*–*k*_EGL_, which reflects the extent of the deviation of the experimentally determined rate of the relaxation from the S_1_ state of the *cis* isomer, from that which can be estimated theoretically based on the energy-gap law. Here, *k_cis_* is the experimentally determined rate of relaxation from the S_1_ state of the *cis* isomer, and *k*_EGL_ is the theoretically predicted rate of the relaxation from the S_1_ state of the *cis* isomer that was derived from the energy-gap law using the values of S_0_–S_1_ energy gap of the *cis* isomer. Therefore, if *k_cis_* − *k*_EGL_ = 0, the experimentally determined S_1_ lifetime of the *cis* isomer strictly obeys energy-gap law, as has been exemplified for the all-*trans* isomer (see solid purple circle in Figure 9). Figure 10 shows the relationship between *k_cis_*–*k*_EGL_ and the S_0_–S_1_ energy gaps of all the isomers investigated in this study. It is interesting that a strong linear relationship was found between *k_cis_*–*k*_EGL_ and the S_0_–S_1_ energy gaps for all the isomers. The result of a least square fitting shows that *k_cis_*–*k*_EGL_ is proportional to the S_0_–S_1_ energy gaps, which clearly demonstrates that the S_1_ lifetimes of the *cis* isomers do not obey the energy-gap law since as the S_0_–S_1_ energy gap increases from that of the all-*trans* isomer the S_1_ lifetime becomes shorter. The S_1_ state lifetimes of the *cis* isomers always deviate from those predicted theoretically, based on the energy-gap law. The deviation is large for the *cis* isomers belonging to Group I (solid red circles and diamonds in Figure 10) and is small for the *cis* isomers belonging to Group II (solid blue circles and diamonds in Figure 10). The S_1_ lifetime of the 7-*cis* isomer (solid green circle in Figure 10) is short compared to the one predicted theoretically, but the datapoint of the 7-*cis* isomer is still along the line of the linear relationship between *k_cis_*–*k*_EGL_ and the S_0_–S_1_ energy gap. The reason behind this systematic linear relationship will be addressed theoretically, using quantum chemical calculations in future study. The present experimental study clearly demonstrates that the S_1_ lifetimes of the *cis* isomers of β-apo-8′-carotenal in *n*-hexane do not obey the well-known energy-gap law for radiationless transitions. Namely, the S_1_ lifetimes of the *cis* isomers become short when the S_0_–S_1_ energy gap becomes large. We suggest that we can call this trend an “inverse energy-gap law”.

### 2.4. Relationship between the S_1_ or S_1_/ICT Lifetimes and the Structures of the Isomers

Table 3 summarizes the lifetimes of the S_1_ or S_1_/ICT species of all the isomers determined by the global analysis. The lifetimes of the S_1_ species are shortened in acetone and methanol due to the production of the S_1_/ICT states in all the isomers compared to those seen in *n*-hexane. This is because the ICT excited state that is coupled to the S_1_ state is stabilized in the polar environment as has already been reported for the all-*trans* isomer [9]. The extent of the stabilization of the S_1_/ICT state is more pronounced in methanol than in acetone judging from the extent of the shortening of the lifetimes.

In order to quantitatively discuss the relationship between the extent of the stabilization of the S_1_/ICT states and the structures of the isomers, we define the quantities *k*_acetone_/*k_n_*_-hexane_ and *k*_methanol_/*k_n_*_-hexane_, where *k_n_*_-hexane_, *k*_acetone_, and *k*_methanol_ shows, respectively, the rate of the relaxation of the S_1_ or S_1_/ICT state of the isomers in *n*-hexane, acetone, and methanol. If *k*_acetone_/*k_n_*_-hexane_ or *k*_methanol_/*k_n_*_-hexane_ becomes large, that means the lifetime of the S_1_/ICT state becomes short and hence the S_1_/ICT state is stabilized in polar solvents (vice versa). Figure 11a shows the relationship between the *k*_acetone_/*k_n_*_-hexane_ and *n_cis_* (the number of C=C bonds from the C=O group to the *cis*-bend in the *cis* isomers as has already been defined above). In acetone, the extent of the stabilization of the S_1_/ICT states of the mono-*cis* isomers (solid red-circles in Figure 11a) is linearly dependent on *n_cis_* except for the 7-*cis* isomer. The di-*cis* isomers (solid blue-circles in Figure 11a) also seem to obey this trend. The extent of the stabilization becomes stronger than the all-*trans* isomer in the case of the isomers whose *n_cis_* is larger than six. Moreover, the stabilization is more pronounced when the *cis*-bend is away from the C=O group. On the other hand, the stabilization becomes weaker than the all-*trans* isomer in the case of the isomers whose *n_cis_* is smaller than five. These findings suggest that the stabilization of the ICT excited states of the *cis* isomers in acetone has an intimate relationship with the distance of the *cis*-bend structure from the C=O group. The anomaly found for the 7-*cis* suggests the presence of an unexpected destabilization of the S_1_/ICT states produced with the 7-*cis* isomer in acetone. We have also tried to correlate the extent of stabilization (*k*_acetone_/*k_n_*_-hexane_) with C6-C8′ distance, but no clear trend was seen. We could only find a linear relationship in the case of taking the distance from the C=O group to the *cis*-bend position into consideration. This finding suggests that the electron withdrawing effect of the carbonyl oxygen may only extend to the nearest *cis*-bend. In other words, the effect of carbonyl oxygen to stabilize the ICT excited state might be interrupted by the presence of the *cis*-bend structure.

On the other hand, in the case of the methanol solution the above situation changes drastically. Figure 11b shows the relationship between the *k*_methanol_/*k_n_*_-hexane_ and *n_cis_*. In methanol, the extent of the stabilization of the S_1_/ICT states of the mono-*cis* isomers (solid red-circles in Figure 11b) is also linearly dependent on *n_cis_* except for the 7-*cis* isomer. The di-*cis* isomers (solid blue-circles in Figure 11b) also seem to obey this trend. The extent of the stabilization always becomes weaker than the all-*trans* isomer except for the 9-*cis* isomer. These findings suggests that there may be additional effects for the destabilization of the S_1_/ICT states of the *cis* isomers in methanol. Nevertheless, a strong linear relationship between *k*_methanol_/*k_n_*_-hexane_ and *n_cis_* holds true for the isomers in methanol, again, except for the 7-*cis* isomer.

Our results suggest that some *cis* isomers, such as the 9-, 13-, and 9,13-*cis* isomers in acetone and the 9-*cis* isomer in methanol, may achieve highly efficient energy-transfer in photosynthetic light-harvesting. This is because these *cis* isomers show a more pronounced stabilization of the S_1_/ICT states than that of the all-*trans* isomer in polar environments. It was shown recently by our group that β-apo-8′-carotenal incorporated into the LH1 light-harvesting system from a carotenoidless, purple, photosynthetic–bacterium *Rhodospirillum rubrum* strain G9+ is indeed in a polar environment, and can act as an efficient accessory light-harvesting pigment [13]. Further investigation is necessary to address this interesting issue.

## 3. Materials and Methods

### 3.1. Sample Preparation of the Isomers of β-Apo-8′-Carotenal

All-*trans* β-apo-8′-carotenal (purity ≥ 96.0%) was purchased from SIGMA and used as it received. Light from W-halogen lamp (50 W) was irradiated for 3 h to *n*-hexane solution of the all-*trans* isomer to generate the *cis* isomers using iodine as a photocatalyst [31]. The *cis* isomers were isolated and purified by the use of a set of HPLC (JASCO PU-4180 and UV-4075). For isolation of the *cis* isomers, three conditions were adopted. Condition 1 (for the isolation of 13-*cis*, 9,13′-*cis*, 9-*cis*, and 7-*cis* isomers): Column, Inertsil SIL-100A 7.6 × 250 mm, 5 µm; Eluent, diethyl ether/*n*-hexane 4/96 *v*/*v*; Flow rate, 2.5 mL/min; Detection wavelength, 450 nm. Condition 2 (for the isolation of the 13′-*cis* and 15-*cis* isomers): Column, LiChrosorb Si 60 7.6 × 300 mm, 5 µm; Eluent, diethyl ether/*n*-hexane 3/97 *v*/*v*; Flow rate, 1.5 mL/min; Detection wavelength, 450 nm. Condition 3 (for the isolation of the 13,13′-*cis* and 9,13-*cis* isomers): Column, Daisopak SP 60 6.0 × 250 mm, 5 µm; Eluent, diethyl ether/*n*-hexane 4/96 *v*/*v*; Flow rate, 1.0 mL/min; Detection wavelength, 450 nm.

### 3.2. Steady-State Absorption Spectroscopy

Steady-state absorption spectra were recorded using a JASCO V-670 spectrophotometer (JASCO Inc., Hachioji, Japan) at room temperature. The purified *cis* isomers were dissolved into three different solvents, *n*-hexane, acetone, and methanol. *n*-Hexane was chosen as a nonpolar solvent, while methanol and acetone were chosen as polar solvents, although acetone is aprotic, while methanol is protic.

### 3.3. Femtosecond Time-Resolved Pump-Probe Absorption Spectroscopy

A part of the output pulses from a femtosecond Ti: Sapphire regenerative amplifier (Spectra Physics, Solstice Ace, 80 fs pulse duration, 3.5 mJ/pulse (3.5 W), 1 kHz repetition) are taken to excite an optical parametric amplifier (Spectra Physics, TOPAS-Prime, Mountain View, CA, USA) and a wavelength converter (Spectra Physics, NirUVis) to generate pump pulses [37]. The central wavelength of the excitation laser pulse was selected where the intensity of absorption becomes 20% of the absorption maximum of the S_0_ → S_2_ transition for each isomer in each solvent in the 490–535 nm spectral region (see Table 4). The intensity of the excitation pulse was set to 10 nJ/pulse and irradiated the samples for 2 h during the measurement. The sample purities were maintained more than 90% after the measurement (see Appendix A). Using a flow cell in time-resolved absorption (TA) measurements is a desirable approach to prevent the accumulation of *trans* structures and their potential effects on the TA data. Appendix A illustrates that when all *cis* isomers absorb light, they undergo a transformation into *trans* structures. This transformation process can result in the gradual accumulation of *trans* structures, which could interfere with the accurate measurement of TA data. However, employing a flow cell requires a significant amount (approximately a few milligrams) of *cis* isomers, making it impractical for highly purified *cis* samples. As an alternative, we utilized a micro-stirrer bar to continuously mix the solution and minimize the effects of isomerization. By doing so, we were able to maintain the isomerization ratio as low as possible (less than 10%). Based on our findings, we believe that the impact caused by isomerization is not highly significant. Another small part of the output from the femtosecond Ti: Sapphire regenerative amplifier (80 fs pulse duration, 100 µJ/pulse (100 mW), 1 kHz repetition) were guided to a sapphire plate, after passing through a rotational-neutral density (ND) filter, to generate the probe super-continuum pulses using the nonlinear optical process of self-phase modulation. A sapphire plate of 2 mm thickness was used to generate the probe pulses when recording the transient absorption spectra in the visible spectra region from 430 to 750 nm, while a second plate with 7 mm thickness was used when recording in the near infrared spectral region from 780 to 1000 nm. The polarization angle between pump and probe pulses was set to 54.7° (the “magic angle”) to avoid the effects caused by the anisotropy. The diameter of the pump beam and probe beam were 200 µm and 250 µm at the sample position. When recording the femtosecond time-resolved absorption spectra, the probe and pump pulses irradiate the sample with a 1 kHz and 500 Hz repetition rate, respectively. The repetition rate of the pump pulses were reduced to 500 Hz using an optical chopper (Terahertz Technologies Inc., Oriskany, NY, USA, C-995 Optical Chopper), synchronously operated with the 1 kHz output signal from the oscillator of the femtosecond regenerative amplifier. The probe pulses that pass through the sample are collected by a spectrometer (Acton Research Corporation, SpectraPro^®^ 275), and the spectral dataset is recorded by a 1024-channel multichannel photodiode (MCPD) detector (Hamamatsu, Linear Image sensor S3903-1024Q mounted on linear image sensor driving circuit C7884). The time-resolution of the present setup is estimated to be around 110 fs by the cross-correlation of pump and probe pulses, supposing both of which have 80 fs pulse duration $\left(80\times \sqrt{2}=113\;\mathrm{fs}\right)$, although the time-width of the instrumental-response function determined by the global analysis is around 60 fs. The accuracy of the steps of the delay-time stage is <10 fs, and the shortest time interval when recording the time resolved absorption spectra was 20 fs. Therefore, when we estimate the standard errors of the rate constants by global analysis, we have adopted 10 fs as the standard error if the estimated error value becomes smaller than 10 fs by the global analysis [13].

The sample solution was set in a quartz cuvette with 2.0 mm optical path-length and was continuously stirred using a micro stirred bar in order to avoid the effect of isomerization during the measurements. The optical density of the sample was set to be around 0.3 at the excitation wavelength. All the time-resolved measurements were performed at room temperature (20–23 °C). The frequency chirp of the femtosecond, time-resolved absorption spectra was corrected experimentally using a 1 mm thickness sapphire plate [37]. The singular value decomposition (SVD) analysis was applied to entire dataset of the observed time-resolved absorption spectra for the purpose of exploring the number of spectrally and temporally independent components in the data matrix. Based on that number of components, a global analysis was applied to obtain the lifetime and evolution associated difference spectra (EADS) of the transient species. The SVD and global analysis were performed using a Glotaran program [38].

### 3.4. Quantum Chemical Calculations

The density functional theory (DFT) calculations were performed to predict the geometry optimized structures of the isomers of β-apo-8′-carotenal in the ground state and in vacuum using a Gaussian 16W program (Version: 64-bit x86-64, Gaussian Inc., Wallingford, USA). The B3LYP/6-31G (d) basis set was used for the computation.

## 4. Conclusions

Our research has produced a set of highly pure *cis* isomers of β-apo-8′-carotenal, and used them to demonstrate three key relationships between their excited-state dynamics. Firstly, we showed that the relationship between the intensity ratio of the *cis*-band/main band and the distance between C6 and C6′ of β-carotene, which was previously proven by calculations, also holds true for the asymmetrical β-apo-8′-carotenal. Secondly, we discovered that the S_1_ state lifetime of the *cis* isomers can be further classified into three types: (13-, 15-) *cis*-bend group (Group I), (13′-, 9-) *cis*-bend group (Group II), and 7-*cis* group (Group III), and proposed an “inverse energy-gap law”, suggesting that the deviation of the rate constant increases linearly with the increase of energy gap between S_0_ and S_1_. Thirdly, we found a correlation between the conjugated length from the carbonyl group to the *cis*-bend and the stabilization of the ICT excited state, providing new insights into the nature of ICT excitation based on the chemical structure. Our results suggest that some *cis* isomers, such as the 9-, 13-, and 9,13-*cis* isomers, could be used to achieve highly efficient energy-transfer in photosynthetic light-harvesting because the S_1_/ICT states produced from these isomers show more pronounced stabilization under a polar environment.

## Figures and Tables

**Figure 1 molecules-28-04424-f001:**
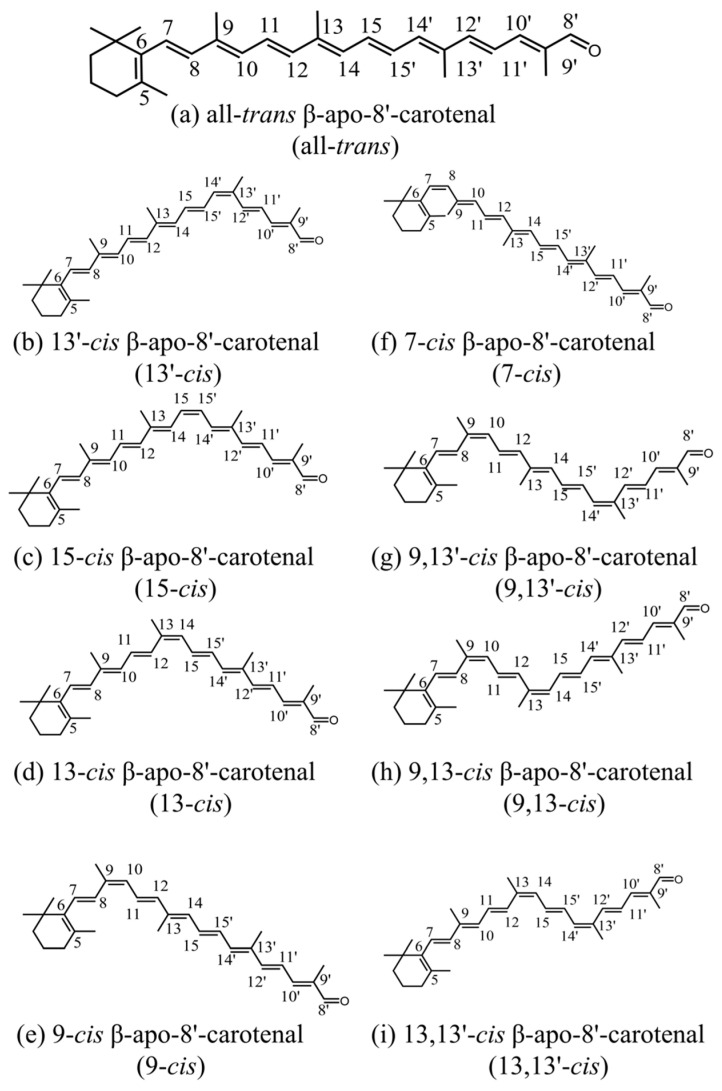
The chemical structures of all-*trans* and *cis* isomers of β-apo-8′-carotenal.

**Figure 2 molecules-28-04424-f002:**
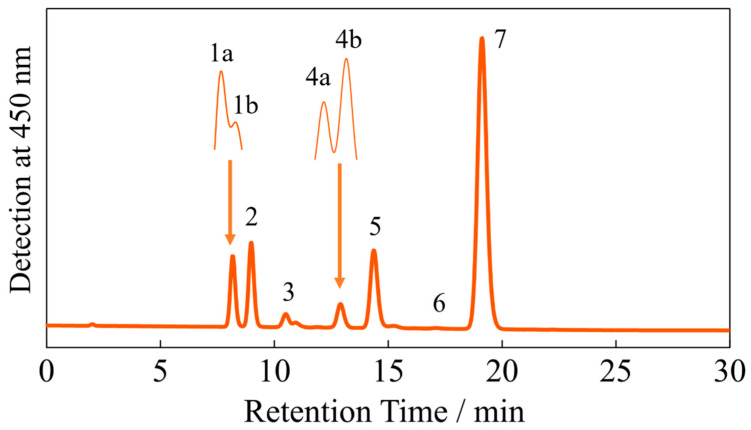
The HPLC profiles detected at 450 nm of the isomers of β-apo-8′-carotenal. The peak numbers indicate the isomers of β-apo-8’-carotenal assigned in Table 1. The inset shows the results of the separation using different column conditions (see Section 3 for the detail).

**Figure 3 molecules-28-04424-f003:**
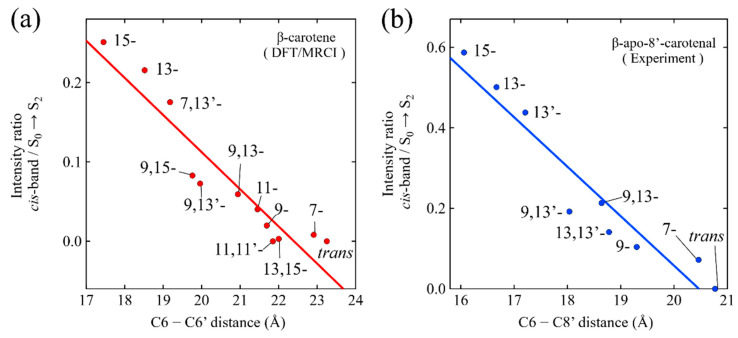
(**a**) Relationship between the C6-C6′ distance of geometrically optimized structures of isomeric β-carotene and the ratio of the theoretically predicted oscillator strength of the *cis*-band transition divided by that of the S_0_ → S_2_ transition of the isomers. This figure was drawn using the data presented in ref. [33]. (**b**) Relationship between the C6-C8′ distance of the geometrically optimized structures of the isomers β-apo-8′-carotenal and the ratio of experimentally determined intensity of the *cis*-band transition divided by that of the S_0_ → S_2_ transition of the isomers in *n*-hexane solutions at room temperature.

**Figure 4 molecules-28-04424-f004:**
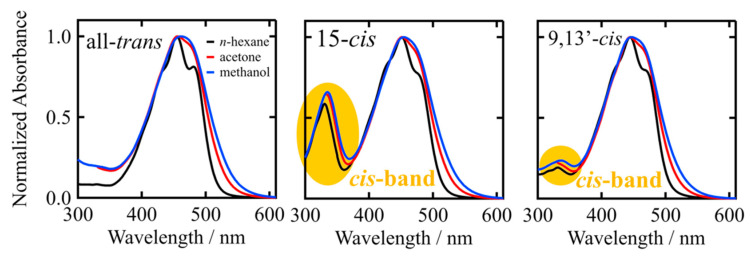
The steady-state absorption spectra of all-*trans*, 15-*cis*, and 9,13′-*cis* isomers of β-apo-8′-carotenal in *n*-hexane, acetone, and methanol at room temperature. The absorption spectra below 330 nm was omitted for acetone solutions since the solvent acetone has strong absorption in this spectral region.

**Figure 5 molecules-28-04424-f005:**
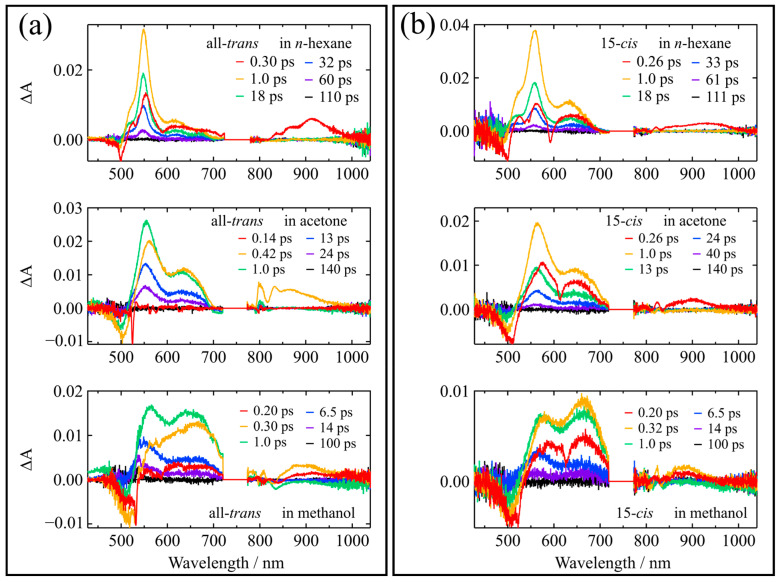
Femtosecond time-resolved absorption spectra of the (**a**) all-*trans* and (**b**) 15-*cis* isomers of β-apo-8′-carotenal in *n*-hexane, acetone, and methanol recorded at selective delay times at room temperature.

**Figure 6 molecules-28-04424-f006:**
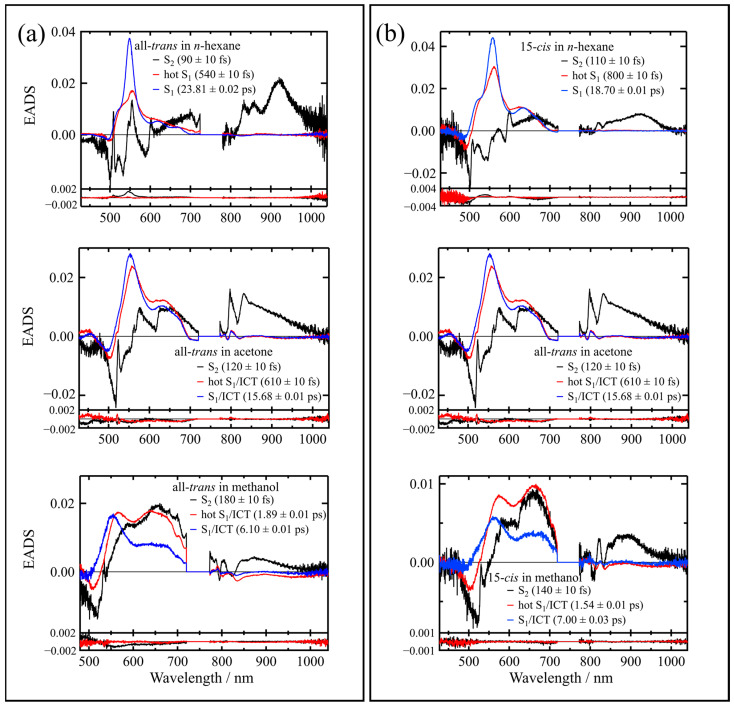
Evolution associated difference spectra (EADS) of the (**a**) all-*trans* and (**b**) 15-*cis* isomers of β-apo-8′-carotenal in *n*-hexane, acetone, and methanol. At the bottom of each panel, the first (black line) and second (red line) right singular vectors of the residual matrix are shown.

**Figure 7 molecules-28-04424-f007:**
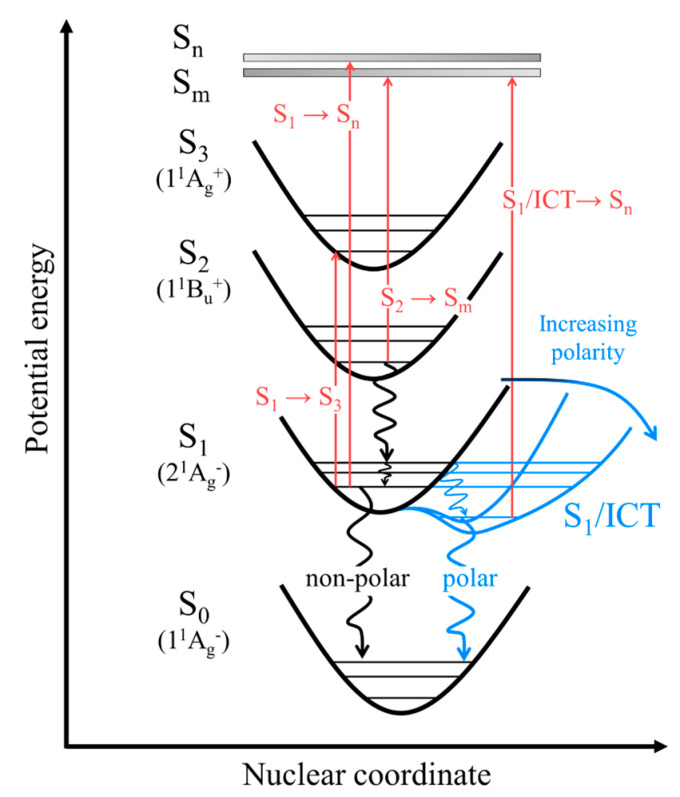
A schematic illustration that shows the potential energy surfaces and optical processes after photoexcitation to the S_2_ state of β-apo-8′-carotenal.

**Figure 8 molecules-28-04424-f008:**
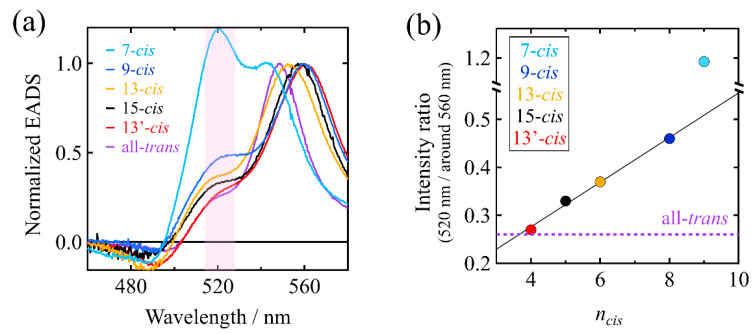
(**a**) Comparison of the normalized EADS of the S_1_ species of the all-*trans* and all the mono-*cis* isomers of β-apo-8′-carotenal in *n*-hexane. The pink shade highlights the difference in the intensity of the 520 nm absorption bands of the isomers. (**b**) The relationship between the intensity ratios 520 nm absorption band/around 560 nm absorption band of the *cis* isomers of β-apo-8′-carotenal and the number of C=C bonds (*n_cis_*) from the C=O group to the *cis*-bend in the *cis* isomers. The intensity ratio of the all-*trans* isomer is shown with a broken, purple line for the reference, as *n_cis_* cannot be defined for the all-*trans* isomer.

**Figure 9 molecules-28-04424-f009:**
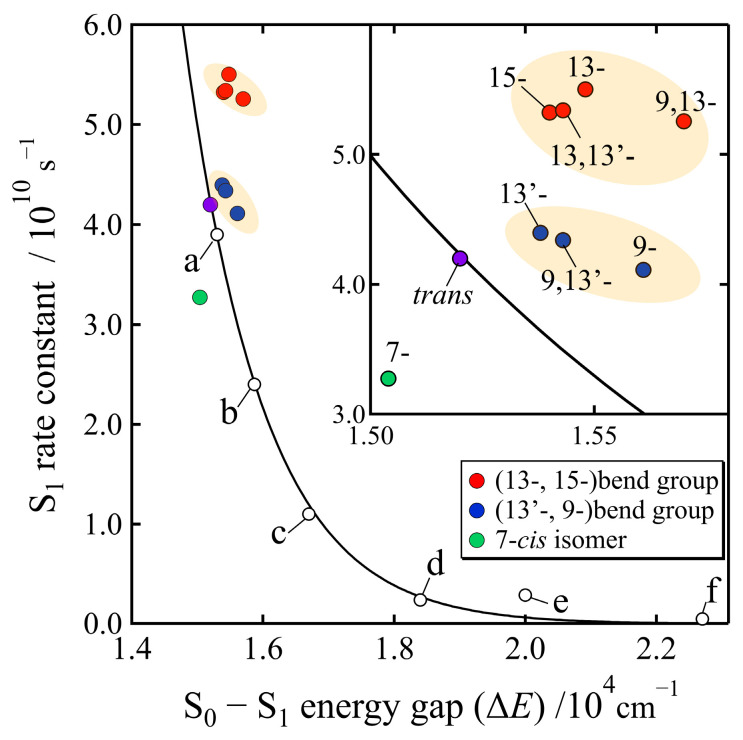
The relationship between the rates of relaxation from the S_1_ state and S_0_–S_1_ energy gaps (∆E) of the isomers of β-apo-8′-carotenal determined in this study and that of the all-*trans* carotenoids reported in the earlier study [34]; a: 3,4-dihydrospheroidene, b: fucoxanthin, c: 3,4,5,6-tetrahydrospheroidene, d: 3,4,7,8-tetrahydrospheroidene, e: mini-7-β-carotene, f: mini-5-β-carotene. The solid black line, which was drawn by the fitting with equations of the energy-gap law (Equations (1) and (2)) to the dataset shown with open black circles, was reproduced from ref. [34]. The inset shows the expansion of the part of the isomers of β-apo-8′-carotenal.

**Figure 10 molecules-28-04424-f010:**
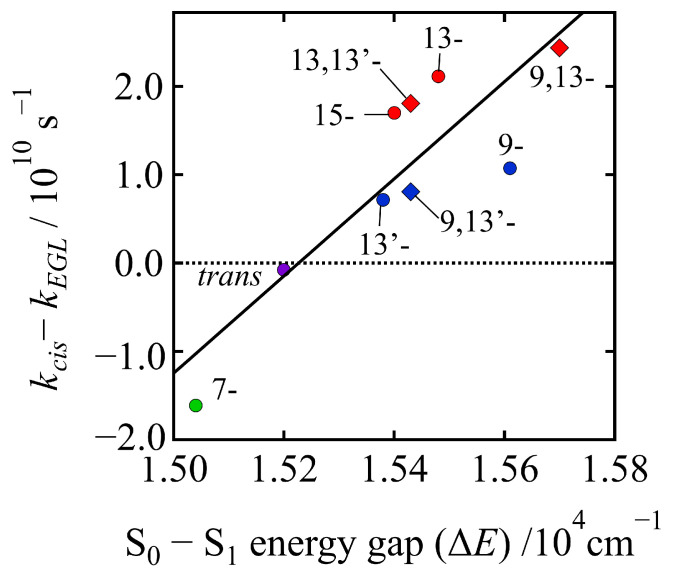
The relationship between *k_cis_*–*k*_EGL_ and the S_0_–S_1_ energy gap of the isomers of β-apo-8′-carotenal in *n*-hexane. Here, *k_cis_* is the experimentally determined rates of relaxation from the S_1_ states of the *cis* isomers, and *k*_EGL_ is the theoretically predicted rates of the relaxation from the S_1_ states of the *cis* isomers that were derived from the energy-gap law using the values of S_0_–S_1_ energy gaps of the isomers. The solid black line was drawn by the least square fitting for all the data points.

**Figure 11 molecules-28-04424-f011:**
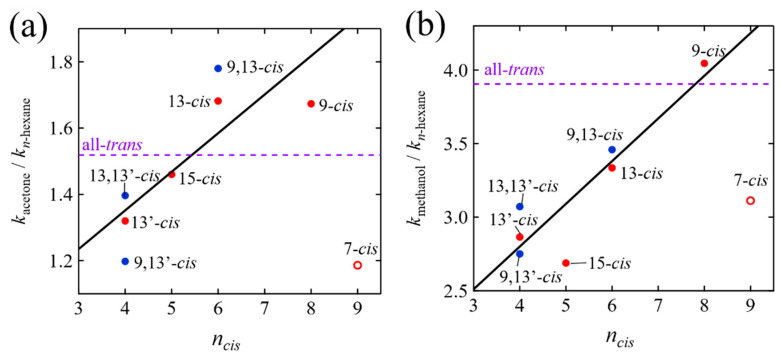
The relationship between the (**a**) *k*_acetone_/*k_n_*_-hexane_ or (**b**) *k*_methanol_/*k_n_*_-hexane_ and the number of C=C bonds (*n_cis_*) from the C=O group to the *cis*-bend in the *cis*-isomers of β-apo-8′-carotenal. Here, *k_n_*_-hexane_, *k*_acetone_, and *k*_methanol_ show, respectively, the rates of the relaxation of the S_1_ or S_1_/ICT states of the isomers in *n*-hexane, acetone, and methanol. The *k*_acetone_/*k_n_*_-hexane_ or *k*_methanol_/*k_n_*_-hexane_ values of the all-*trans* isomer were illustrated with a broken purple line for the reference. The solid black lines were drawn by the least square fittings of the data without the 7-*cis* isomer.

**Table 1 molecules-28-04424-t001:** The assignment of the peaks in the chromatogram shown in Figure 2 and the absorption wavelengths of the isomers of β-apo-8′-carotenal in *n*-hexane. The absorption wavelengths are determined by the second-derivative waveforms of the absorption spectra. The value in parenthesis shows the wavelength shift with reference to the all-*trans* isomer.

		*cis*-Band (nm)		S_0_ → S_2_ (nm)		
Peak Number	Isomer	0-1	0-0	0-2	0-1	0-0
1_a_	13′-*cis*	318	332	420 (8)	448 (7)	478 (8)
1_b_	15-*cis*	315	331	424 (4)	452 (3)	483 (3)
2	13-*cis*	316	332	423 (5)	449 (6)	478 (8)
3	9,13′-*cis*	316	331	418 (10)	446 (9)	476 (10)
4_a_	13,13′-*cis*	-	334	415 (13)	442 (13)	472 (14)
4_b_	9,13-*cis*	315	330	419 (9)	446 (9)	474 (12)
5	9-*cis*	-	329	424 (4)	451 (4)	481 (5)
6	7-*cis*	-	331	430 (−2)	453 (2)	483 (3)
7	all-*trans*	-	-	428 (−)	455 (−)	486 (−)

**Table 2 molecules-28-04424-t002:** The wavelengths of the S_1_ → S_n_ or S_1_/ICT → S_n_ absorption maxima of the isomers of β-apo-8′-carotenal. The parenthesis indicates the wavelength shift with reference to the all-*trans* isomer.

Isomer	*n*-Hexane (nm)	Acetone (nm)	Methanol (nm)
7-*cis*	520 (−27)	547 (−6)	549 (−4)
all-*trans*	549 (0)	553 (0)	553 (0)
13-*cis*	553 (4)	557 (4)	561 (8)
15-*cis*	558 (9)	562 (9)	562 (9)
9-*cis*	559 (10)	564 (11)	564 (11)
13′-*cis*	561 (12)	564 (11)	562 (9)
9,13-*cis*	561 (12)	566 (13)	562 (9)
13,13′-*cis*	562 (13)	565 (12)	564 (11)
9,13′-*cis*	564 (15)	569 (16)	568 (15)

**Table 3 molecules-28-04424-t003:** The lifetimes of the S_1_ or S_1_/ICT species of isomeric β-apo-8′-carotenal in *n*-hexane, acetone, and methanol at room temperature.

Isomer	*n*-Hexane (ps)	Acetone (ps)	Methanol (ps)
all-*trans*	23.81 ± 0.02	15.68 ± 0.01	6.10 ± 0.01
7-*cis*	30.55 ± 0.02	25.76 ± 0.02	9.82 ± 0.02
9-*cis*	24.32 ± 0.01	14.53 ± 0.01	6.01 ± 0.02
13-*cis*	18.18 ± 0.01	10.81 ± 0.01	5.45 ± 0.02
15-*cis*	18.79 ± 0.02	12.87 ± 0.01	6.99 ± 0.03
13′-*cis*	22.67 ± 0.01	17.18 ± 0.02	7.91 ± 0.01
9,13-*cis*	19.03 ± 0.01	10.69 ± 0.02	5.50 ± 0.02
13,13′-*cis*	18.73 ± 0.01	13.41 ± 0.01	7.71 ± 0.02
9,13′-*cis*	23.04 ± 0.02	19.23 ± 0.02	8.37 ± 0.01

**Table 4 molecules-28-04424-t004:** The central excitation wavelength (nm) of the excitation laser pulse used for the femtosecond time-resolved absorption spectroscopy on the isomers of β-apo-8′-carotenal in *n*-hexane, acetone, and methanol.

Isomer	*n*-Hexane	Acetone	Methanol
all-*trans*	510	525	535
7-*cis*	500	515	525
9-*cis*	505	520	530
13-*cis*	500	515	525
15-*cis*	505	520	530
13′-*cis*	505	515	525
9,13′-*cis*	500	510	520
9,13-*cis*	495	510	520
13,13′-*cis*	490	505	515

## Data Availability

The data that support the findings of this study are available from the authors on reasonable request.

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
