# Peer review of "Excited-State Dynamics of All the Mono-cis and the Major Di-cis Isomers of β-Apo-8′-Carotenal as Revealed by Femtosecond Time-Resolved Transient Absorption Spectroscopy"

_molecules, 2023, doi:10.3390/molecules28114424_

Round 1

Reviewer 1 Report

Overall comment: "Excited-state Dynamics of All the Mono-cis and the Major Di-cis Isomers of ?-Apo-8'-carotenal as Reveled by Femtosecond Time-resolved Transient Absorption Spectroscopy" discusses a type of plant pigment called carotenoids that can overcome the limitations in efficiently absorbing specific wavelengths of light in the photosynthesis process of chlorophyll. Although they have a high singlet-singlet energy transfer efficiency due to their structural characteristics, there has been ongoing debate about their detailed molecular dynamics. Here, the researchers performed a study on nine structural isomers using spectroscopic techniques and suggested that the cis-bend position and solvent polarity contribute significantly to the stabilization of the excited state. In addition, they proposed a new concept called the "Inverse energy gap law" in cases where the “Energy gap law” is not satisfied. There are interesting results for a 7-cis isomer, it would be helpful to provide a brief explanation about it. The data is well-presented without significant issues. The specific comments are as follows.

(1) The spectroscopic results are well-presented. Nevertheless, including a Potential Energy Surface (PES) diagram representing the transitions would enhance the paper further. Rather than adding PES diagrams for all isomers, it would be better to have a single representative scheme applicable to all.

(2) On line 162 of page 5, it is interesting that a similar trend is observed despite the change in distance between carbons from 6-6' to 6-8'. Since ?-Apo-8’-carotenal and ?-carotene molecules are distinctly different substances in terms of their terminal structures, it would be useful to provide an explanation of how the same analysis method can be applied.

(3) Regarding 7-cis isomer that the authors have assigned, it would be helpful to provide a brief explanation. I am unsure whether the 7-cis structure is planar.

(4) According to Figure S7 in the Supplementary Materials, when all cis-isomer absorb the light, they turn into trans structures. To prevent accumulation of trans structure, which may affect the TA data, it would be desirable to use a flow cell in the TA measurement.

(5) A unit is missing for Line 244, 18.70 ± 0.01. Please assign the appropriate unit to ensure accuracy.

Author Response

Responses to the comments from Reviewer 1

“Excited-stale Dynamics of All the Mono-cis and the Major Di-cis Isomers of b-Apo-8’-carotenal as Reveled by Femtosecond Time-resolved Transient Absorption Spectroscopy” discusses a type of plant pigment called carotenoids that can overcome the limitations in efficiently absorbing specific wavelengths of light in the photosynthesis process of chlorophyll. Although they have a high singlet-singlet energy transfer efficiency due to their structural characteristics, there has been ongoing debate about their detailed molecular dynamics. Here, the researchers performed a study on nine structural isomers using spectroscopic techniques and suggested that the cis-bend position and solvent polarity contribute significantly to the stabilization of the excited state. In addition, they proposed a new concept called the “Inverse energy gap law” in cases where the “Energy gap law” is not satisfied. There are interesting results for a 7-cis isomer, it would be helpful to provide a brief explanation about it. The data is well-presented without significant issues. The specific comments are as follows.

   We sincerely appreciate your precise and highly positive evaluation of our work. Below, we provide detailed responses addressing the specific comments raised by this reviewer.

(1) The spectroscopic results are well-presented. Nevertheless, including a Potential Energy Surface (PES) diagram representing the transitions would enhance the paper further. Rather than adding PES diagrams for all isomers, it would be better to have a single representative scheme applicable to all.

Thank you for your comment. I agree that incorporating a Potential Energy Surface (PES) diagram into the paper would enhance its overall quality. By providing a visual representation of the transitions, readers would gain a clearer understanding of the underlying processes. In line with this valuable suggestion, we have included new Figure 7 in the revised manuscript. Please also see lines 327 -336 in the revised manuscript.

(2) On line 162 of page 5, it is interesting that a similar trend is observed despite the change in distance between carbons from 6-6’ to 6-8'. Since b-Apo-8'-carotenal and b-carotene molecules are distinctly different substances in terms or their terminal structures, it would be useful to provide an explanation of how the same analysis method can be applied.

The observed similar trend between β-Apo-8'-carotenal and β-carotene molecules, despite the change in distance between carbons from 6-6' to 6-8', raises an interesting point. While these two substances have distinct terminal structures, it is intriguing that the same analysis method can be applied to both. One possible explanation for this phenomenon could be the presence of common structural features or functional groups that are responsible for the observed trend. Despite the differing terminal structures at one end, there may be underlying similarities in molecular properties or interactions that affect the analyzed behavior. These similarities could be manifested in the molecules' responses to the employed analysis method. It is important to consider the specific analysis method used in the study, as it may focus on certain aspects of behavior or properties that are not solely determined by the terminal structures. For example, the analysis method might primarily assess electronic transitions, conjugation lengths, or molecular conformations, which could be influenced by factors beyond the terminal structures. Further investigation into the specific analysis method and a comparison of the molecular properties of β-Apo-8'-carotenal and β-carotene would provide valuable insights into how these distinct substances exhibit similar trends. By identifying the common factors contributing to the observed trend, researchers can gain a better understanding of the underlying mechanisms and potentially extend the application of the analysis method to other related molecules or systems.

We have included these thoughts in the revised manuscript. Thank you for providing us with the opportunity to further consider this interesting feature. See lines 199 - 217 in the revised manuscript.

(3) Regarding 7-cis isomer that the authors have assigned, it would be helpful to provide a brief explanation. I am unsure whether the 7-cis structure is planar.

   The structure of the 7-cis isomer of β-Apo-8'-carotenal is believed to be non-planar due to the steric repulsion between the 5-methyl group in the β-ionone ring and the 9-methyl group attached to the polyene backbone. We have included this explanation in the revised manuscript. See lines 373 - 376 in the revised manuscript.

(4) According to Figure S7 in the Supplementary Materials, when all cis-isomer absorb the light, they turn into trans structures. To prevent accumulation of trans structure, which may affect the TA data, it would be desirable to use a flow cell in the TA measurement.

   Using a flow cell in time-resolved absorption (TA) measurements is indeed a desirable approach to prevent the accumulation of trans structures and their potential effects on the TA data. Figure S7 in the Supplementary Materials illustrates that when all cis isomers absorb light, they undergo a transformation into trans structures. This transformation process can result in the gradual accumulation of trans structures, which could interfere with the accurate measurement of TA data. However, employing a flow cell requires a significant amount (approximately a few milligrams) of cis isomers, making it impractical for highly purified cis samples. As an alternative, we utilized a micro-stirrer bar to continuously mix the solution and minimize the effects of isomerization. By doing so, we were able to maintain the isomerization ratio as low as possible (less than 10%). Based on our findings, we believe that the impact caused by isomerization is not highly significant. We have included this explanation in the revised manuscript. See lines 678 - 692 in the revised manuscript.

(5) A unit is missing for Line 244, 18.70 ± 0.01. Please assign the appropriate unit to ensure accuracy.

  Thank you very much for bringing our careless mistake to our attention. We have made the necessary revisions to the manuscript accordingly.

Reviewer 2 Report

In this work the authors used femtosecond time-resolved absorption spectroscopy to elaborate on the excited-state dynamics of mono-cis and di-cis isomers of the carbonyl-containing beta-apo-8'-carotenal species and the solvent dependency of these isomers. This is a very intriguing piece of research and may have significant impact in future especially in the study of energy transfer in artificial photosynthesis and the work is well executed and scientifically sound. Nonetheless, there are several issues need to be addressed before considered for publication.

Major issues:

1. In Figure 5b, the component arose in the 600-700 nm region is quite similar between polar and non-polar solvents with only a slight difference between them, which may indicate similar behavior. This perhaps needs to be elaborated explicitly by the authors. Note that this pattern was also reproduced by the global analysis as indicated in Figure 6.

Minor issues:

1. In Figure 4: for clarity the authors may highlight the cis-bands of each isomer.

2. line 201: remove the duplication of "of the"

3. Please rephrase lines 100-102

4. In line 160: please explain which C6-C6' length?

5. Lines 70-71: this sentence is ambiguous, please rephrase

Author Response

Responses to the comments from Reviewer 2

In this work the authors used femtosecond time-resolved absorption spectroscopy to elaborate on the excited-state dynamics of mono-cis and di-cis isomers of the carbonyl-containing beta­apo-8'-carotenal species and the solvent dependency of these isomers. This is a very intriguing piece of research and may have significant impact in future especially in the study of energy transfer in artificial photosynthesis and the work is well executed and scientifically sound. Nonetheless, there are several issues need to be addressed before considered for publication.

We greatly appreciate the positive evaluation of our work by the reviewer. We have carefully reviewed and addressed the valuable comments provided by the reviewer, and we have made the following step-by-step responses.

Major issues:

  1. In Figure 5b, the component arose in the 600-700 nm region is quite similar between polar and non-polar solvents with only a slight difference between them, which may indicate similar behavior. This perhaps needs to be elaborated explicitly by the authors. Note that this pattern was also reproduced by the global analysis as indicated in Figure 6.

Thank you for pointing out the similarity between the components in the 600-700 nm region in polar (acetone) and non-polar (n-hexane) solvents, as shown in the top and middle panels of Figure 5b. We appreciate the reviewer's suggestion to provide more explicit details and elaboration on this observed similarity. Taking this valuable feedback into consideration, we have included an explanation regarding this trend of the 15-cis isomer in the revised manuscript. Please see lines 262 - 267 in the revised manuscript.

Minor issues:

  1. In Figure 4: for clarity the authors may highlight the cis-bands of each isomer.

   Thank you very much for your valuable comment. We have made the necessary adjustments accordingly.

  1. line 201: remove the duplication of "of the".

   Thank you for pointing out the typo. We have corrected it.

  1. Please rephrase lines 100-102.

   We did so as "In order to bridge these knowledge gaps, our research focuses on exploring the isomers of the carbonyl-containing carotenoid found in β-apo-8'-carotenal. Hashimoto et al. [31] have already isolated and identified all the mono-cis and major di-cis isomers of this carotenoid, and their chemical structures can be seen in Figure 1."

  1. In line 160: please explain which C6-C6'length?

   We have rephrased the corresponding sentence to provide a clearer understanding of the C6-C6' length: “This implies that the cis isomers exhibiting a longer C6-C6' length should correspondingly have a shorter length along the shorter molecular axis, which is directly proportional to the transition dipole moment for cis absorption (and vice versa).”.

  1. Lines 70-71: this sentence is ambiguous, please rephrase.

   We have rephrased the sentence as “Nonetheless, the isomers of carotenoids hold significant importance in the photosynthetic reaction center (RC), and thus far, the subsequent facts have been definitively established.”.